# Treatment of Statistical Estimation Problems in Randomized Smoothing for Adversarial Robustness

**Václav Voráček**
Tübingen AI center, University of Tübingen
`vaclav.voracek@uni-tuebingen.de`

## Abstract

Randomized smoothing is a popular certified defense against adversarial attacks. In its essence, we need to solve a problem of statistical estimation which is usually very time-consuming since we need to perform numerous (usually $10^5$) forward passes of the classifier for every point to be certified. In this paper, we review the statistical estimation problems for randomized smoothing to find out if the computational burden is necessary. In particular, we consider the (standard) task of adversarial robustness where we need to decide if a point is robust at a certain radius or not using as few samples as possible while maintaining statistical guarantees. We present estimation procedures employing confidence sequences enjoying the same statistical guarantees as the standard methods, with the optimal sample complexities for the estimation task and empirically demonstrate their good performance. Additionally, we provide a randomized version of Clopper-Pearson confidence intervals resulting in strictly stronger certificates. The code can be found at `https://github.com/vvoracek/RS_conf_seq`.

We encourage the readers only interested in statistics to start at Subsection 2.1.

## 1 Introduction

**Adversarial robustness:** It is well known that a tiny, adversarial, perturbation of the input can change the output of basically any undefended machine learning model (Biggio et al., 2013; Szegedy et al., 2014); this is unpleasant and we continue in the mitigation of the problem. There are two main lines of work tackling this problem: (1) Empirical: the standard approach here is to use adversarial training (Madry et al., 2018; Goodfellow et al., 2014) where the model is trained on adversarial examples. This approach does not provide guarantees, only empirical evidence suggesting that the model may be robust. With stronger attacks, we might (yet again) realize it is not the case. (2) Certified: with formal robustness guarantees for the model. We will focus on this, and in particular on *randomized smoothing* (Lecuyer et al., 2019) which is currently the strongest certification method[1]. We will not cover other certification methods and we refer the reader to the survey Li et al. (2023) instead. We consider the standard task of certified robustness; the goal is to decide if the decision of a classifier $F$ at a particular input $x$ is robust against additive perturbations $\delta$ such that $\|\delta\| \leq r$ for some norm $\|\cdot\|$. Formally, we ask if $F(x) = F(x')$ whenever $\|x - x'\| \leq r$.

**Randomized smoothing** is a framework providing state of the art formal guarantees on the adversarial robustness for many datasets. One of its benefits lies in the fact that there are no assumptions on the model, making it possible to readily transfer the methods from defending image classifiers against sparse pixel changes to different modalities; e.g., defending large language models against change of

---

[1]see leaderboard `https://sokcertifiedrobustness.github.io/leaderboard/`

38th Conference on Neural Information Processing Systems (NeurIPS 2024).

some letters/words/tokens. Randomized smoothing transforms any undefended classifier $F : \mathbb{R}^d \to \mathcal{Y}$ by a smoothing distribution $\varphi$ into a smoothed classifier $H_\varphi(x) = \arg\max_{y \in \mathcal{Y}} \mathbb{P}_{\delta \sim \varphi}[\![ F(x + \delta) = y ]\!]$ for which robustness guarantees exist. We postpone the details for later. During the certification process, we need to estimate the maximum probability of a multinomial distribution from samples as the exact computation is intractable. This statistical estimation problem is the focus of this paper.

**Speed issues:** The main weakness of randomized smoothing is the extensive time required for both prediction and certification, making it troublesome for real-world applications. There is an inherent trade-off between the allowed probability of incorrectly claiming robustness[2] (type-1 error, $\alpha$), the probability of incorrectly claiming non-robustness (type-2 error, $\beta$), and the number of samples used $n$. The standard practice is to set $\alpha = 0.001$, $n = 100\,000$ and the value of $\beta$ is then implicit. it might not be the most practically relevant setting since the implicitly set value of $\beta$ is exponentially small in $n$ when the sample is not close to the threshold. The claim is made precise in Example A.1.

The arguably more relevant setting is to set the values of $\alpha$ and $\beta$ and leave $n$ implicit. This is much more challenging since it is no longer possible to draw the predetermined number of samples and use a favourite concentration inequality. We propose a new certification procedure using confidence sequences to adaptively (and optimally) deciding how many samples to draw addressing the problem.

**Contributions:**

- We introduce a new, strictly better version of Clopper-Pearson confidence intervals for estimating the class probabilities in Subsection 2.1. The presented interval is optimal, and thus is the ultimate solution to the canonical statistical estimation of randomized smoothing.

- We propose new methods for the certification utilizing confidence sequences (instead of confidence intervals) in Subsection 2.2. This allows us to draw *just enough* samples to certify robustness of a point; greatly decreasing the number of samples needed.

- We provide a complete theoretical analysis of the proposed certification procedures. In particular, we provide matching (up to a constant factor) lower-bounds and upper-bounds for the width of the respective confidence intervals. We invert the bound and show that the certification procedure has the optimal sample-complexity in an adaptive estimation task.

- We provide empirical validation of the proposed methods confirming the theory.

**Notation:** Bernoulli random variable with mean $p$ is denoted as $\mathcal{B}(p)$ and binomial random variable is $\mathcal{B}(n, p)$. Random variables are in capitals ($X$) and the realizations are lowercase ($x$). We type sequences in bold and denote $\mathbf{x}_{:t}$ first $t$ elements of $\mathbf{x}$. We write $a \lesssim b$ if there exists a universal constant $C > 0$ such that $a \leq Cb$. If $a \lesssim b$ and $b \lesssim a$, then we write $a \asymp b$. Iverson bracket $[\![ \Phi ]\!]$ evaluates to 1 if $\Phi$ is true and to 0 otherwise.

## 1.1 Paper organization

First, in Section 2 we introduce randomized smoothing, then, in Subsection 2.1, we introduce Clopper-Pearson confidence intervals, show that they are conservative and propose their improved (optimal) randomized version. In Subsection 2.2 we discuss shortcomings of confidence intervals and introduce confidence sequences and provide lower and upper bounds for their performance. We use the confidence sequences in Section 3 and benchmark them on a sequential estimation task.

## 2 Randomized Smoothing

As outlined in Introduction, consider a classifier $F : \mathbb{R}^d \to \mathcal{Y}$ and let the class probabilities under additive noise $\varphi$ be $h_\varphi(x)_y = \mathbb{P}_{\delta \sim \varphi}[F(x + \delta) = y]$. Denote the highest probability (breaking ties arbitrarily) class in the original point $A = \arg\max_{y \in \mathcal{Y}} h_\varphi(x)_y$ and the second-highest probability class $B = \arg\max_{y \in \mathcal{Y} \setminus A} h_\varphi(x)_y$. Let the corresponding probabilities be $p_A$ and $p_B$ respectively. Recalling that $H_\varphi(x) = \arg\max_{y \in \mathcal{Y}} h_\varphi(x)_y$, then for a certain function $r : [0, 1]^2 \to \mathbb{R}_+$ we have

$$\|x - x'\| \leq r(p_A, p_B) \implies H_\varphi(x) = H_\varphi(x').$$

---

[2]of an input for a model at a certain radius

This $r$ depends on the smoothing distribution $\varphi$ and the considered norm. For example, if the considered norm is $\ell_2$ and $\varphi$ is isotropic Gaussian with standard deviation $\sigma$, then $r(p_A, p_B) = \frac{\sigma}{2}(\Phi^{-1}(p_A) - \Phi^{-1}(p_B))$ where $\Phi^{-1}$ is Gaussian quantile function Cohen et al. (2019). Note that in general, $r(\cdot, \cdot)$ is increasing in the first coordinate and decreasing in the second one. The intuition is that the larger the value of $p_A$ at $x$, the larger it will be also in the neighborhood of $x$; similarly for $p_B$. It is common in the literature to use the bound $p_B \leq 1 - p_A$ and thus certify $r(p_A, 1 - p_A)$. We stick to the convention in the paper and discuss the topic in more details in Appendix B.

**Statistical estimation:** The crux of the paper lies in the statistical estimation problems for randomized smoothing. We consider the abstract framework for randomized smoothing, so the proposed techniques can be used as a drop-in replacement in all randomized smoothing works with a statistical-estimation component (i.e., not in the de-randomized ones such as Levine & Feizi (2021)). We do not only propose methods that work good empirically, we also provide theoretical analysis suggesting that we solve the problems optimally in a certain strong sense. The main focus is on the following two constructs.

1. Confidence intervals: A standard component of randomized smoothing pipelines is the Clopper-Pearsons confidence interval. It is known to be conservative[3]; thus, the certification procedures are unnecessarily underestimating the certified robustness. We provide the *optimal* confidence interval for binomial random variables, resolving this issue completely.

2. Confidence sequences: In the standard randomized smoothing practice, we draw a certain, predetermined, number of samples and then we compute the certified radius on a confidence level $1 - \alpha$. We improve on this by allowing for adaptive estimation procedures employing confidence sequences; We demonstrate the performance in the standard task of adversarial robustness, where we want to decide if a point is robust at radius $r$ with type-1 (resp. 2) error rates $\alpha$ (resp. $\beta$) using as few samples as possible.

**Literature review of randomized smoothing:** The most relevant related works are Horváth et al. (2022); Chen et al. (2022) and they are discussed in Section 3. Here we briefly summarize literature relevant to randomized smoothing in general. The choice of the smoothing distribution $\varphi$ is a crucial decision determined mainly by the threat model with respect to which we want to be robust. For example, if we are after certifying $\ell_1$ robustness, we choose a uniform distribution in a $d-$dimensional $\ell_\infty$ ball (Lee et al., 2019; Yang et al., 2020), or better, splitting noise (Levine & Feizi, 2021), but we do not go into details here. Alternatively, for $p-$norms, $p \geq 2$ one would usually use a $d-$dimensional normal distribution Lecuyer et al. (2019); Cohen et al. (2019). The variance of the distribution based on how large perturbations do we allow in our threat model. We refer the reader to Yang et al. (2020) for a broader discussion on the smoothing distributions. It is possible to use methods in the spirit of randomized smoothing to certify other threat models, such as patch attacks Levine & Feizi (2020) and sparse attack Bojchevski et al. (2020), which can be readily extended to other modalities. Sometimes, the "smoothing" distribution can be made supported on a small, discrete set and then we can evaluate the expectation exactly, yielding deterministic certification (often called de-randomized smoothing (Levine & Feizi, 2021)). See also Kumari et al. (2023) for a survey on randomized smoothing containing examples of when the certification is beyond additive $\ell_p$-norm threat model; even such techniques use the Bernoulli estimation subroutine.

## 2.1 Confidence Intervals

We do not have access to the class $A$ probability $p_A$ and only have to estimate it from binomial samples; hence, the name *randomized* smoothing. Because of the randomness, we can only provide probabilistic statements about the robustness of a classifier in the following spirit "with probability at least $1 - \alpha$, robust radius is at least $r$" for a small $\alpha$, usually $0.001$. This failure probability corresponds to the event of overestimating $p_A$ and we control it with the help of confidence intervals.

The standard choice for calculating the upper confidence interval is the Clopper-Pearson interval, sometimes called *exact*. Regardless of this pseudonym, it is in reality conservative. In this subsection, we introduce the Clopper-Pearson confidence interval for the mean of binomial random variables, demonstrate its limitations and introduce its (better) randomized version.

---

[3]See `https://en.wikipedia.org/wiki/Binomial_proportion_confidence_interval`.

**Definition 2.1** (Confidence interval for binomials). Let $u, v$ map sample to a real number. They form a (possibly randomized) confidence interval $I(x) = [u(x), v(x)]$ with coverage $1 - \alpha$ if for any $p \in [0, 1]$ it holds that

$$\mathbb{P}_{X \sim \mathcal{B}(n,p), I}\left(p \in I(X)\right) \geq 1 - \alpha.$$

We will mainly use one-sided confidence intervals; that is, $u(\cdot) = 0$ (lower confidence interval) or $v(\cdot) = 1$ (upper confidence interval). When we will talk about probability of confidence interval containing the parameter, it will be in the frequentist sense, keeping in mind that confidence intervals provide no guarantees post-hoc for any individual estimation.

Clearly, if $I(x) = [0, 1]$ regardless of $x$, it will be a valid confidence interval but rather useless; thus we aim for short intervals. Ideally it would hold for every $p$ that $\mathbb{P}_X(p \notin I(X)) = \alpha$, otherwise, some values are included in the confidence intervals unnecessarily often they can be shortened. In the following we introduce the standard Clopper-Pearson confidence intervals Clopper & Pearson (1934).

**Definition 2.2** (Clopper-Pearson intervals). One sided upper interval is defined as $v(x) = 1$ and

$$u(x) = \inf\{p \mid \mathbb{P}(\mathcal{B}(n, p) \geq x) > \alpha\}.$$

The lower one is defined as $u(x) = 0$ and

$$v(x) = \sup\{p \mid \mathbb{P}(\mathcal{B}(n, p) \leq x) > \alpha\}.$$

Amongst the deterministic confidence intervals, they are the shortest possible; however, they are in general conservative. In the binomial case $\mathcal{B}(n, p)$, there are only $n + 1$ possible outcomes; and thus only $n + 1$ possible confidence intervals suggesting that the actual coverage can be $1 - \alpha$ only for at most $n + 1$ values of $p$. The problem strikingly arises for upper confidence interval for large values of $p$. When we sample from $\mathcal{B}(n, p)$, regardless of the outcome, all values larger than $\sqrt[r]{\alpha}$ are contained in the confidence interval. This is a usual problem in the context of randomized smoothing, leading to sharp drops towards the end of robustness curves. We demonstrate this sub-optimality in the first part of Example A.2. We mitigate this problem by introducing randomness into the confidence intervals. They will still have the desired coverage level $1 - \alpha$, but will be shorter. Intuitively, we do so by "interpolating" between the deterministic confidence intervals in the spirit of Stevens (1950).

**Definition 2.3** (Randomized Clopper-Pearson intervals). Let $W$ be uniform on the interval $[0, 1]$. The randomized one sided upper interval is defined as $v_r(x) = 1$ $u_r(x) = u'_r(x, W)$ where

$$u'_r(x, w) = \inf\{p \mid \mathbb{P}(\mathcal{B}(n, p) > x) + w\mathbb{P}(\mathcal{B}(n, p) = x) > \alpha\}.$$

The lower one is defined as $u_r(x) = 0$ and $v_r(x) = v'_r(x, W)$ where

$$v'_r(x, w) = \sup\{p \mid \mathbb{P}(\mathcal{B}(n, p) < x) + w\mathbb{P}(\mathcal{B}(n, p) = x) > \alpha\}.$$

**Proposition 2.4.** *Randomized Clopper-Pearson interval ($I_{rCP}$) have coverage exactly $1 - \alpha$. Furthermore, for any confidence interval $I$ at level $1 - \alpha$, and any $p \geq q \in [0, 1]$ it holds that*

$$\mathbb{P}_{X \sim \mathcal{B}(n,p)}(q \in I(X)) \geq \mathbb{P}_{X \sim \mathcal{B}(n,p)}(q \in I_{rCP}(X)).$$

The proof is in the Appendix D and we remark that the interval can be efficiently found by binary search. Proposition 2.4 implies that the randomized Clopper-Pearson bounds are optimal and all the other confidence intervals for binomial random variables are more conservative. It remains to demonstrate the advantage of the randomized confidence intervals. We refer to Figure 1 for the comparison of the randomized and deterministic Clopper-Pearson confidence intervals and how they affect the robustness. We note that the most significant difference is towards the high values of $p$ and for certification functions $r$ such that $\lim_{p \to 1} r(p) = \infty$, such as when smoothing with normal distribution. This explains the common sharp drop by the end of the robustness curve.

**Width of confidence intervals:** For the simplicity of exposition, let the width of a confidence interval at level $1 - \alpha$ with $n$ samples be $\asymp \sqrt{\log(1/\alpha)/n}$. This way, we hide the dependency on $p$ into $\asymp$. In the full generality, the width of the confidence intervals exhibits many decay regimes between the rates $\sqrt{p(1-p)\log(1/\alpha)/n}$ (when $np(1-p) \gg 1$) and $\log(1/\alpha)/n$ (when $np(1-p) \asymp 1$). Our algorithms capture the correct scaling of the confidence intervals. See Boucheron et al. (2013) and the discussion on Bennett's inequality which captures the correct rates for Bernoulli mean estimation.

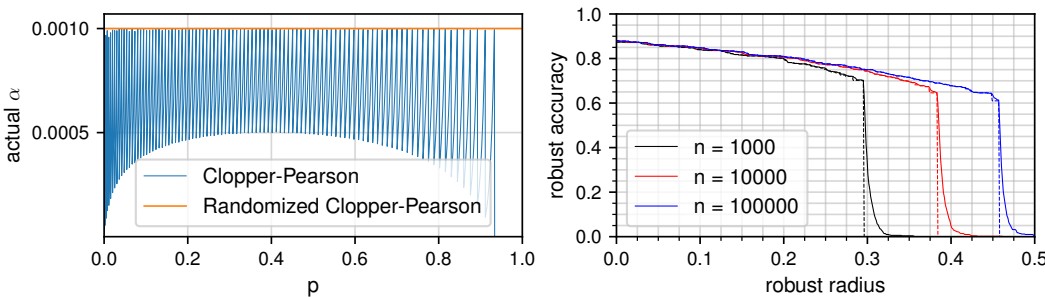

Figure 1: **left**: Comparison of coverages of confidence intervals for the mean estimation of $\mathcal{B}(100, p)$ when $\alpha := 0.001$. Note that for $p > \sqrt[n]{\alpha} \sim 0.93$, the coverage is 1. **right**: Comparison of $\ell_2$-robustness curves with the standard (dashed) or the randomized (solid) Clopper-Pearson bounds on a CIFAR-10 dataset under the standard setting. The experimental details are in Appendix C.

## 2.2 Confidence Sequences

**Limitations of confidence intervals:** In order to compute the confidence intervals presented in the previous subsection, we need to collect samples and then run an estimation procedure once which brings certain limitations. Consider the following two scenarios: (1) It might be the case that we do not need all 100 000 samples and after only 10 it would be enough for our purposes because we could already conclude that the point cannot be certified here; thus, we wasted 99 990 samples. (2) Alternatively, we could see that even $10^5$ samples are not enough, and we need to draw more samples. However, we have already spent our failure budget $\alpha$, so we cannot even carry another test at all.

This motivates the introduction of confidence sequences. They generalize confidence intervals in the way that they provide a confidence interval after every received sample such that we control the probability that the underlying parameter is contained in *all* the confidence intervals *simultaneously*.

**Definition 2.5** (Confidence sequence). Let $\{u_t, v_t\}_{t=1}^{\infty}$ be mappings from a sequence of observations to a real number. They form a confidence sequence $I_t(\mathbf{x}_{:t}) = [u_t(\mathbf{x}_{:t}), v_t(\mathbf{x}_{:t})]$ for all $t \geq 1$ with confidence level $1 - \alpha$ if

$$\mathbb{P}\left(p \in I_t(\mathbf{X}_{:t}), \forall t > 1\right) \geq 1 - \alpha$$

for any $p \in [0, 1]$, where $\mathbf{X}$ is an infinite sequence of Bernoulli random variables $\mathcal{B}(p)$.

*Remark* 2.6. Since we want the estimated parameter to be contained in all the confidence intervals simultaneously, we will have by convention that $I_{t+1}(x_{:t+1}) \subseteq I_t(x_{:t})$.

To simplify presentation, we would consider the symmetric ones; i.e., those where we consider the two possible failures - when we overestimate or underestimate the mean - to be equal. However, they will be constructed from two one-sided bounds, so the generalization is straightforward and will not be discussed.

**Related work:** The construction of confidence sequences based on union bound employs the doubling trick which is widely used in online learning to convert fixed-horizon algorithms to anytime algorithms. In this direction, we refer to Mnih et al. (2008) as the direct predecessor of this work, where they used similar techniques but did not explicitly construct confidence sequences. In the confidence sequence literature, this technique is similar to stitching of Howard et al. (2020). The stitched confidence intervals of Howard et al. (2020) are generally shorter by a small constant factor, but the analysis and generalization become complicated, contrasting with our approach.

The construction based on betting is in the spirit of Orabona & Jun (2023) (the reference contains an excellent survey on the topic). The difference mainly lies in the fact that we are interested in Bernoulli random variables, which allows us to use specialized tools at places, as opposed to the referred work, which considers bounded random variables. As an analogy to confidence intervals, the previous work constructed Bernstein-type inequalities, while we constructed Clopper-Pearson-like bounds. In Ryu & Bhatt (2024), the authors improved over Orabona & Jun (2023) in certain aspects in the case of $[0, 1]$-valued random variables. Notably, in Section 3, they considered the special case - $\{0, 1\}$-valued random variables and their results are greatly overlapping those in Section 2.4.

## 2.3 Union bound confidence sequence

A natural way how to extend the confidence intervals to confidence sequences is to construct a confidence interval at every time step and use a union bound to control the total failure probability. In the following, we first show that a naive application of this approach is asymptotically suboptimal, and then we provide a way how to construct optimal confidence intervals in a certain strong sense.

**Intuition on the width of confidence sequences:** For any random variable with finite variance, the optimal width of the confidence interval for the mean parameter scales as $\sqrt{\log(1/\alpha)/t}$ with the increasing number of samples $t$ at confidence level $1 - \alpha$ Lugosi & Mendelson (2019). On the other hand, it is well known that the width of the optimal confidence sequence scales as $\sqrt{(\log(1/\alpha) + \log\log t)/t}$ as $t$ increases due to the law of iterated logarithms Ledoux & Talagrand (1991). A naive use of union bound, computing a confidence interval using failure probability at time step $t$, $\alpha_t = \frac{\alpha c}{t^\gamma}$ for some $c$ and $\gamma > 1$ such that $\sum_{i=1}^{\infty} \alpha_t = \alpha$ yields a confidence sequence whose width scales as $\sqrt{\log(1/\alpha_t)/t} \approx \sqrt{(\log(t) + \log(1/\alpha))/t}$. We cannot choose any monotonous $\alpha_t$ schedule decaying slower because even for $\gamma = 1$ we still keep the $\log$ factor while the sum $\sum_{i=1}^{\infty} \delta_t$ diverges.

Now consider non-monotonous schedules of $\alpha_t$, two key ideas follows. (1) In order to have the optimal rate $\log(1/\alpha_t) \approx \log(1/\alpha) + \log\log t$, we need $\alpha_t \asymp \alpha/\log t$. Clearly, if this holds for all $t$, then $\sum_{t=1}^{\infty} \alpha_t$ diverges. (2) A confidence interval at time $t$ is also a valid confidence interval for all $t' > t$. Furthermore, if $t'$ is not much larger than $t$, then it may still asymptotically have the optimal width up to a multiplicative constant. Thus, updating the confidence sequence when $t$ is a power of (say) 2 result in the optimal width. This reasoning is formalized in the following theorem.

**Theorem 2.7.** *Fix $\alpha > 0$. Consider a sequence*

$$\alpha_t = \begin{cases} \frac{\alpha}{k(k+1)} & \text{if } t = 2^k \text{ for integer } k, \\ 0 & \text{otherwise.} \end{cases}$$

*Then Algorithm 1 produces a confidence sequence at level $1 - \alpha$ of the following width which is attained in the worst case*

$$\varepsilon_t \lesssim \sqrt{\frac{\log(1/\alpha) + \log\log(t)}{t}}$$

*where $\varepsilon_t$ is $U - L$ at time $t$, and the confidence intervals are randomized Clopper-Pearson intervals.*

The proof is in Appendix E. We remark that the statement of Theorem 2.7 prioritizes simplicity over its full generality. The generalization to other schedules of $\alpha_t$ from the second bullet point as described in the previous paragraph is routine and described in detail in Appendix C.2.

**Corollary 2.8.** *The asymptotic rate of 2.7 is optimal due to law-of-iterated-logarithm (Ledoux & Talagrand, 1991) and even in the finite sample regime due to Balsubramani (2014)[Theorem 2].*

## 2.4 Confidence sequences based on betting

A recent alternative approach to confidence sequences is based on a hypothetical betting game. For the illustration, consider a fair sequential game; e.g., sequentially betting on outcomes of a coin. If we guess the outcome correctly, we win the staked amount, otherwise we lose it. If the coin is fair, in expectation, our wealth stays the same. On the other hand, if the game is not fair and the coin is biased, we can win money. For example, if the true head-probability is $0.51$, we start increasing our wealth in an exponential fashion, see Example A.3; thus, if we win lots of money, we can conclude that the game is not fair. We instantiate a betting game for every possible mean $0 \leq p \leq 1$ that would be fair if the true mean is $p$. Then we observe samples of the random variable and as soon as we win enough money, we drop that particular $p$ from the confidence sequence. To make things formal, we introduce the necessary concepts from probability theory. The evolution of our wealth throughout a fair game is modeled by martingales[4], sequences of random variables for which, independently of the past, the expected value stays the same.

---

[4]It would be historically accurate to say that martingales actually model fair games.

**Definition 2.9** (Martingale). A sequence of random variables $W_1, W_2, \ldots$ is called a *martingale* if for any integer $n > 0$, we have $\mathbb{E}(|W_n|) < \infty$ and $\mathbb{E}(W_{n+1}|W_1, \ldots, W_n) = W_n$. If we instead have $\mathbb{E}(W_{n+1}|W_1, \ldots, W_n) \leq W_n$, then the sequence is called a *supermartingale*.

| **Algorithm 1** Union-Bound Confidence Sequence | **Algorithm 2** Betting Confidence Sequence |
|---|---|
| $t, H, K, L, U \leftarrow 0, 0, 0, 0, 1$ 
 **loop** 
    Obtain random $x$ 
    $H \leftarrow H + x$ 
    $t \leftarrow t + 1$ 
    **if** $t = 2^K$ **then** 
       $K \leftarrow K + 1$ 
       $\alpha_t \leftarrow \alpha/(K(K+1))$ 
       $L \leftarrow \max\{L, \text{LowConfInt}(H, t, \alpha_t)\}$ 
       $U \leftarrow \min\{U, \text{UppConfInt}(H, t, \alpha_t)\}$ 
    **end if** 
 **end loop** | $\text{LOGQ}, t, H, L, U \leftarrow 0, 0, 0, 0, 1$ 
 **loop** 
    $\hat{q} \leftarrow (H + 1/2)/(t + 1)$ 
    Obtain random $x$ 
    $H \leftarrow H + x$ 
    $t \leftarrow t + 1$ 
    $\text{LOGQ} \leftarrow \text{LOGQ} + x \log(\hat{q}) + (1-x)\log(1-\hat{q})$ 
    $\text{LOGP}(p) := H \log(p) + (t - H)\log(1 - p)$ 
    $I_p \leftarrow \{p | \text{LOGQ} - \text{LOGP}(p) \leq \log(1/\alpha)\}$ 
    $L \leftarrow \max\{L, \min I_p\}$ 
    $U \leftarrow \min\{U, \max I_p\}$ 
 **end loop** |

In the coin-betting example, $W_1, W_2, \ldots$ is a martingale where $W_n$ represents our wealth after playing the game for $n$ rounds. We stress that $W_t \geq 0$ for all $t > 0$. By convention, we will also have $W_1 = 1$. We further need a time-uniform generalization of Markov's inequality.

**Proposition 2.10** (Ville's inequality Durrett (2010)). *Let $W_1, W_2, \ldots$ be a non-negative supermartingale. then for any real $a > 0$*

$$\mathrm{P}\left[\sup_{n \geq 1} W_n \geq a\right] \leq \frac{\mathbb{E}[W_1]}{a}.$$

Thus, whenever we play a game and earn a lot, we can — with high probability — rule out the possibility that the game is fair. So far, this is still an abstract framework. We have yet to design the betting game and the betting strategy and describe how to run the infinite number of games.

**Betting game:** Let[5] $0 < p < 1$. Consider a coin-betting game where we win $1/p$ (resp. $1/(1-p)$) multiple of the staked amount if we correctly predicted heads (resp. tails). If the underlying heads probability is $p$, then regardless of our bet - in expectation - we still have the same amount of money; thus, this game is fair. We identify heads and tails with outcomes $1, 0$ respectively.

**Betting strategy:** We deconstruct the betting strategy into the two sub-tasks: (1) If we know the underlying heads probability, we can design the optimal betting strategy for any criterion. (2) Estimate the heads probability. **First sub-task:** Let $p$ define the betting game from the previous paragraph and $q$ be the true heads probability; Optimally, bet $q$-fraction of wealth to heads and $1 - q$ fraction to tails. It maximizes the expected log-wealth, or equivalently, the expected growth-rate of our wealth and is also known as the Kelly Criterion. This is known to be optimal for the adaptive estimation task, see Wald (1947) under the name sequential-probability-ratio-test (SPRT). One might have expected the optimal criterion to optimize to be the expected wealth; however, the betting strategy maximizing the expected wealth suggest to bet all the money on one of the outcomes. This strategy, however, leads to an eventual bankruptcy almost surely and is not recommended in this context. **Second sub-task:** A natural choice is to use the running sample mean of the observations as the estimator of $q$. Unfortunately, this estimator would be either $0$ or $1$ after the first observations, so we go bankrupt whenever the observed sequence contains both outcomes. Thus, we use a "regularized" sample mean and after observing $H$ times heads in a sequence of length $t$, we estimate $\hat{q} = (H + 0.5)/(t + 1)$. This is the MAP estimate of the mean with $\text{Beta}(1/2, 1/2)$ prior and is known as Krichevsky–Trofimov estimator (Cesa-Bianchi & Lugosi (2006) Section 9.7), Krichevsky & Trofimov (1981) and is proven to be successful for building confidence sequences beyond the Bernoulli case Orabona & Jun (2023).

**Parallel betting games:** We have described a betting game for a certain $p$ and a betting strategy. Employing Ville's inequality we can possibly reject the hypothesis that the true sampling distribution

---

[5]For simplicity of exposition, similar arguments hold when $p \in \{0, 1\}$.

follows $p$. However, we need to run the game for all values of $p \in [0,1]$ and after every observation report the smallest interval containing all the values of $p$ that were not rejected so far. This is clearly impossible to do explicitly, but it turns out that the non-rejected values of $p$ form an interval and we can use binary search twice to find the end-points of the interval. This is a non-trivial result and generally does not need to hold for confidence intervals constructed by betting. The first key observations is that the betting strategy does not depend on $p$, so we can "play the game" just once. The second observation is that the resulting wealth is convex in $p$. To see why, let $\hat{q}_\tau$ (resp. $x_\tau$) be our estimate of $q$ (resp. the coin-toss outcome, for brevity $1/0$ corresponds to heads/tails respectively and $H = \sum_{\tau=1}^{t} x_\tau$), then our log-wealth at time $t$ can be written as a function of $p$.

$$\log W_t(p) = \log \prod_{\tau=1}^{t} \left( \left( \frac{\hat{q}_\tau}{p} \right)^{x_\tau} \left( \frac{1 - \hat{q}_\tau}{1 - p} \right)^{1 - x_\tau} \right)$$

$$= \underbrace{\sum_{\tau=1}^{t} x_\tau \log(\hat{q}_\tau) + (1 - x_\tau) \log(1 - \hat{q}_\tau)}_{\text{LOGQ}} \underbrace{-H \log(p) - (t - H) \log(1 - p)}_{\text{LOGP}(p)}.$$

Therefore, at every time-step, we can compute the interval of values of $p$ for which the betting game has not concluded yet and thus form the current confidence interval. The proof is in Appendix F.

**Theorem 2.11.** *Algorithm 2 produces a valid confidence sequence at confidence level $1 - \alpha$, where we interpret the interval $[L, U]$ at iteration $T$ of the algorithm as the confidence interval at time $t$ with width $\varepsilon$ at most (which is attained in the worst case):*

$$\varepsilon_t \lesssim \sqrt{\frac{\log(1/\alpha) + \log(t)}{t}}.$$

This is not asymptotically optimal; still, empirically it performs well, and the same techniques can be used to obtain a confidence sequence that follows law-of-iterated-logarithms Orabona & Jun (2023). We present a comparison of the confidence sequences in Figure 2 and conclude this subsection by an implementation remark.

*Remark 2.12.* Wealth $W(\mathbf{x}_{:t})$ does not depend on the order of $\mathbf{x}_{:t}$, so we write it as $W(h, t)$, meaning wealth after observing $h$ heads in the first $t$ tosses. Now, for every time $t$, we can compute what is the minimal number of observed 1 (heads) (call it $H(t)$) so that $p$ is outside of the lower-confidence interval. $H(t)$ is clearly non-decreasing in $t$; also, $W(h, t)$) can be easily computed from $W(h - 1, t)$ and from $W(h, t-1)$ in constant time. The whole dynamic programming approach can be summarized in the following scheme which is repeatedly executed starting from $h = 0, t = 0$.

- If $W(h, t) \geq \frac{1}{\alpha}$: $H(t) := h$, $t := t + 1$, compute $W(h, t + 1)$
- Else: $h := h + 1$, compute $W(h + 1, t)$.

Both lines are executed in constant time. Also, we start from $W(0, 0)$ and $h, t < N + 1$. Executing a line, either $h$ or $t$ increases and thus to compute the thresholds for sequence of length up to $N$, we need to execute at most $2N$ lines of the scheme. We note that for the simplicity, we did not handle the case when $W(h, t) < 1/\alpha$. In that case we would just set $H(t) = t + 1$ and increase $t$.

## 3 Experiments

Now we shall provide experimental evidence for the performance of the proposed methods. We benchmark the confidence sequences on the Sequential decision making task, where we try to certify a certain radius at given confidence level with as few samples as possible; the definition that follows is general beyond randomized smoothing. We emphasize that this setting of certifying a certain radius is by far the most common one in the robustness literature. We stress that the comparison of the robustness curves (e.g., as in Figure 1) is vacuous, since in the adaptive task, we do not spend samples to *improve* the robustness curves, beyond the certified level.

**Definition 3.1** (Sequential decision making task). Let $\frac{1}{2} \leq p, q \leq 1$ and only $p$ being known. Receive samples from $\mathcal{B}(q)$. After every sample, either halt and declare that $p > q$, $p < q$, or request another sample. The task is to minimize the number of samples while being wrong with frequency at-most $\alpha$.

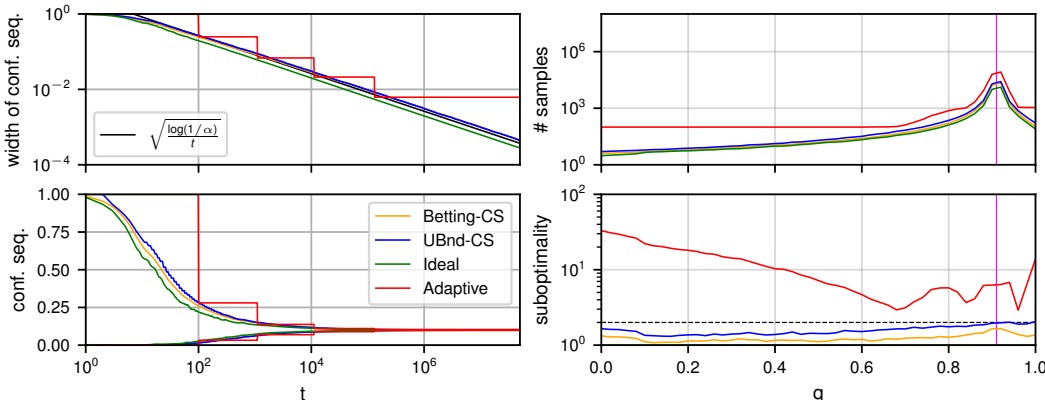

Figure 2: **left:** Comparison of widths of confidence sequences for the mean of Bernoulli $\mathcal{B}(0.1)$ with $\alpha = 0.001$. The width is on top and the actual confidence sequence on the bottom. In the notation of Algorithms 1 and 2, the sequence of $U - L$ is in the top figure, while both sequences $U$ and $L$ are in the bottom figure. Note the log-scale for $t$ (and width on top). **right** : Instantiation of Task 3.1. The goal is to decide if $p = 0.91$ (vertical magenta line) or not with $\alpha = 0.001$. On top are the numbers of samples requested for the individual methods averaged over 1000 trials for 51 equally spaced values of $p \in [0, 1]$; on the bottom is the relative suboptimality of the individual methods; i.e., how many times more samples did they request compared to the ideal method. Note log scales on the $y-$axis. **methods:** UBnd-CS and Betting-CS are from Algorithm 1 and 2 respectively. Adaptive is from Horváth et al. (2022). The ideal is the unattainable lower-bound for the two tasks. On the LHS, it is a confidence interval on level $1 - \alpha$ computed independently at every time step. On the RHS, it is SPRT knowing both $p, q$ which is optimal due to Wald (1947).

## 3.1   Related work

We identified Horváth et al. (2022) as the most relevant work. They distinguish between samples for which a predetermined radius $r$ can be certified, and the samples for which it cannot. They use $s$ values $n_1 < \cdots < n_s$ ($[10^2, 10^3, 10^4, 1.2 \cdot 10^5]$) sequentially as the number of samples. They try to certify radius $r$ with $n_1$ samples; if it fails, then they try $n_2$ samples etc. They employ Bonferroni correction (union bound) and every sub-certification is allowed to fail with probability only $\frac{\alpha}{s}$. The key differences (details in Appendix C.2.1) to our method are that (1) It always abstains for hard tasks. (2) Splitting the $\alpha$ budget evenly degrades performance for small $n$. (3) method is only a heuristics. See Figure 5 and Tables 1, 3, 4. For the empirical comparison.

Another relevant work is Chen et al. (2022). Here, the certification is split in two phases. (1) Mean is crudely estimated. (2) The crude estimate selects the number of samples drawn so that the decrease (either multiplicative or absolute) in the certified radius is heuristically approximately at most a predetermined constant. We note that this heuristic for distributing samples can be made rigorous in a certain sense (see Dagum et al. (1995)). This is trivial for confidence sequences, as one can stop the estimation only as soon as they short enough and solve the task of Chen et al. (2022) with guarantees (instead of just heuristic). In this sense, we see our methods to be more general. We benchmark this in Table 2.

The works Seferis et al. (2023); Ugare et al. (2024) also address the speed issues of randomized smoothing, however, they are orthogonal to our directions. In particular, Ugare et al. (2024) uses an auxiliary network for which the certification is faster and transfer the certificates to the original model. Seferis et al. (2023) observes that few samples are sufficient for non-trivial certificates.

## 4   Conclusion

In this paper, we investigated the statistical estimation procedures related to randomized smoothing and improved them in the following two ways: (1) We have provided a strictly stronger version of confidence intervals than the Clopper-Pearson confidence interval. (2) We have developed confidence sequences for sequential estimation in the framework of randomized smoothing, which will greatly

|                              | r=0.5        | r=1.25 ,       | r=2          |
|------------------------------|--------------|----------------|--------------|
| Adaptive Horváth et al. (2022) | $1976 \pm 41$ | $3593 \pm 574$ | $4623 \pm 47$ |
| Betting CS 2                 | $531 \pm 157$ | $2169 \pm 257$ | $2130 \pm 339$ |
| Union bound CS 1             | $635 \pm 157$ | $2557 \pm 234$ | $2670 \pm 271$ |
| Adaptive Horváth et al. (2022) | $0.13 \pm 0.006$s | $0.23 \pm 0.036$s | $0.3 \pm 0.003$s |
| Betting CS 2                 | $0.05 \pm 0.012$s | $0.17 \pm 0.019$s | $0.17 \pm 0.02$s |
| Union bound CS 1             | $0.05 \pm 0.006$s | $0.19 \pm 0.018$s | $0.21 \pm 0.02$s |

Table 1: Comparison of the average number of samples (resp. time) needed to decide if a point is certifiably robust with given radius. Cifar10, $\ell_2$, details are in Appendix C.2.1

| $\varepsilon$ | 0.01    | 0.02   | 0.03   |
|---------------|---------|--------|--------|
| UB-CS         | 197 628 | 49 198 | 21 513 |
| Betting-CS    | 199 771 | 47 215 | 20 918 |
| Horvath       | 768 560 | 94 900 | 81 080 |

Table 2: We run the confidence sequences until the width is smaller than $\varepsilon$ on a (both sided) confidence level 0.999. That way we can certify certain radius knowing that the true probability is at most $\varepsilon$ larger. We used the same network as for the $\ell_2$ experiment in Table 1 (WideResnet-40 on CIFAR10, $\sigma = 1$). We report the average number of samples required over 500 images.

reduce the number of samples needed for adaptive estimation tasks. Additionally, we provided matching algorithmic upper bounds with problem lower bounds for the relevant statistical estimation task.

## 5  Broader Impact Statement

We hope that this paper enlarges the interest in statistical estimation within the ML community.

## Acknowledgments

The author was supported by the DFG Cluster of Excellence "Machine Learning – New Perspectives for Science", EXC 2064/1, project number 390727645 and is thankful for the support of Open Philanthropy.

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

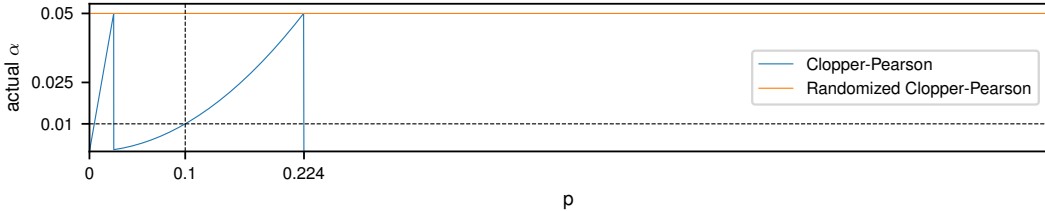

Figure 3: Actual coverages for (randomized) Clopper-Pearson confidence intervals for $\mathcal{B}(2, p)$.

## A Deferred examples

**Example A.1.** [implicit type-2 error is exponentially small] Let us sample from $X \sim \mathcal{B}(p)$ to decide if the mean is smaller or larger than $p - \varepsilon$ using $n = 100\,000$ samples. We use Hoeffdings' inequality to bound the probability that $p$ is incorrectly estimated to be lower than $p - \varepsilon$, i.e.,

$$\mathbb{P}\left[\frac{1}{n}\sum_{i=1}^{n} X_i \leq \mathbb{E}[X] - \varepsilon\right] \leq e^{-2n\varepsilon^2}.$$

Considering $\varepsilon$ to be a constant, we see that this probability scales as $e^{-n}$. For example, when the true probability is $0.5$ and we want to decide if it is smaller or larger than $0.4$, already $1000$ samples make the probability of incorrectly decision to be roughly $2 \cdot 10^{-9}$.

**Example A.2.** [suboptimality of Clopper-Pearson confidence interval and optimality of the randomized one] Recall that the coverage for $p$ is the probability that $p$ is included in the confidence interval when it is the true parameter and $\alpha$ is the allowed type-1 error and $1 - \alpha$ should be the coverage. Consider samples from $X \sim \mathcal{B}(2, p)$ and $\alpha = 0.05$. By definition, the Clopper-Pearson upper intervals are $[0, 1], [0.025, 1], [0.224, 1]$ for observations $x = 0, 1, 2$ respectively. Coverage for $p = 0.224$ is $0.95$ because the event that $p$ is outside of the confidence interval is $\mathbb{P}(X \in \{0, 1\}) = 1 - p^2 \approx 0.95$. On the other hand, coverage for $p = 0.1$ is $\mathbb{P}(X \in \{0, 1\}) = 1 - p^2 = 0.99$ and for all $p > 0.224$ it is $1$; see Figure A for the coverages. Now we turn on to the randomized Clopper-Pearson interval for $p = 0.5$. Recall the definition of the upper interval,

$$u'_r(x, w) = \inf\{p \mid \mathbb{P}(\mathcal{B}(n, p) > x) + w\mathbb{P}(\mathcal{B}(n, p) = x) > \alpha\}.$$

The randomized confidence interval for some value $x$ interpolates between the confidence intervals for $x$ and $x + 1$ when $x < n$. Thus, when $x \neq 2$, $p > 0.224$ is always in the confidence interval (this happens with probability $1 - p^2 = 0.75$. Otherwise, we solve the following for $w$ (because we set $n = 2$, $x = 2$):

$$\mathbb{P}(\mathcal{B}(2, p) > 2) + w\mathbb{P}(\mathcal{B}(2, p) = 2) = \alpha,$$
$$0 + p^2 w = \alpha,$$
$$v = \alpha/p^2.$$

Thus, $p > 0.224$ is not contained in the randomized confidence interval iff $W \leq \alpha/p^2$ and $X = 2$. Since these random variables are independent, the resulting probability is $\mathbb{P}(W \leq 0.2)\mathbb{P}(X = 2) = \alpha/p^2 \cdot p^2 = \alpha$, as desired.

**Example A.3** (Exponential increase of wealth for a biased coin)**.** For simplicity of exposition we assume that the coin falls on head $51$ times from $100$ tosses. To get a high probability statement is straightforward. Let us always bet $0.51$ fraction of our money to to heads and $0.49$ to tails (equivalently, just bet $0.02$ of the money to the heads). If we win, we win $2\%$ of our money, otherwise we lose $2\%$. Then, our wealth after $100$ tosses will be $1.02^{51} \cdot 1.02^{-49} \sim 1.04$. Thus, every $100$ tosses we multiply our wealth by the factor of $1.04$ which is the desired exponential function.

## B Binary or multiclass certification

Although all the standard benchmarking datasets for randomized smoothing are multiclass (cifar10 and imagenet), the commonly used randomized smoothing certification protocol is for the binary

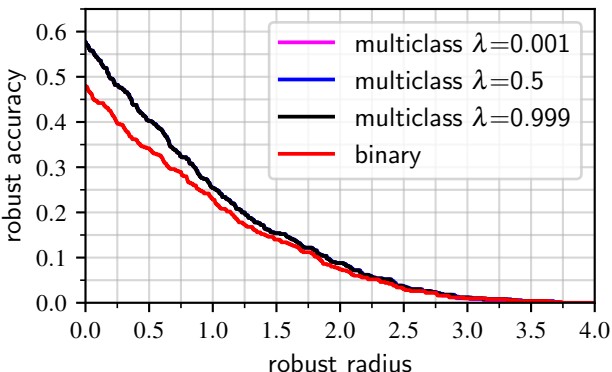

Figure 4: Comparison of the robustness curves for binary and multiclass certification. In the binary case, all the failure budget $\alpha = 0.001$ was spent on controlling the top-1 class probability. In the multiclass setting, we spend $\lambda$ fraction of the budget in bounding $p_A$ and the remaining $1 - \lambda$ part on bounding $p_B$. Note that this has no significant effect. The average certified radius for binary certification is $0.50$, while for the multiclass it is $0.61$. The experimental details are in Appendix C. The only difference is that now $\sigma = 1$.

setting, where we certify class $A$ against all the other classes merged in a super class. In that case, the certification is done using the formula $r(\hat{p}_A, 1 - \hat{p}_A)$, where we have to guarantee that $p_A \geq \hat{p}_A$ at confidence level at least $1 - \alpha$. The alternative is to use multiclass certification; here, the certification is done via formula $r(\hat{p}_A, \hat{p}_B)$ ensuring that $p_A \geq \hat{p}_A$ and $p_B \leq \hat{p}_B$ at the same time at confidence level at least $1 - \alpha$. The difference between these two bounds naturally manifests in the regime when $p_A$ is small. Strikingly, when $p_A < 0.5$, the binary certification approach cannot certify any robustness, while the multiclass one possible can. The cost for the multiclass procedure is only that we have to divide the failure budget between the the two estimation procedures. This is usually insignificant. In the $\ell_2$ (and thus $\ell_\infty$ case), the role of $\alpha$ is rather minor, see Cohen et al. (2019) Figure 8. This is even more pronounced in the $\ell_1$ case. Here $\alpha$ plays an absolutely negligible role in the resulting certified radius, see Voracek & Hein (2023), Subsection 2.7 for the discussion and Figures 4, 6. While this might be known to many, we believe that some readers may benefit from reading this argument. We demonstrate the difference in certification power in Figure 4. This multiclass certification fits in our setting effortlessly. We can run one confidence sequence for $p_A$, and another for $p_B$. We do not even need to know $p_B$ and we can run it for all of them at the same time. This means, that we run it only for the second most observed class. This second most observed class does not need to be the actual runner-up class, but since it was possibly observed more times than the actual runner-up class, it will also provide a wider confidence interval, so the statistical estimation is still correct.

## C    Experimental details

### C.1    Figure 1

The model in Figure 1 is the pretrained cifar10 model (Exactly the same model/setting from the example in README) of Salman et al. (2019), `https://github.com/Hadisalman/smoothing-adversarial`; in particular, it was ResNet-110 smoothed with Gaussian noise $\sigma = 0.12$ for $\ell_2$ robustness. We set $\alpha = 0.001$ as usual and skip every 20 images of the test dataset (using 500 images, as is the standard practice).

### C.2    Parameters of union bound confidence sequences

We were enlarging the sample size by a factor of $\beta = 1.1$ between estimations (that is, the condition $T = 2^K$ is replaced by $T > \beta^K = 1.1^K$ and our schedule is $\alpha_k = 5\alpha/((t+4)(t+5))$) The method is not sensitive to the choice of hyperparameters, see 5. In fact, the hyperparameters $\beta, \gamma$ should be selected based on what is the "interesting" regime. There is an inherent tradeoff between a good asymptotical performance ($\beta, \gamma \sim 1$) and low-sample performance ($\beta, \gamma > 1$). This is confirmed in

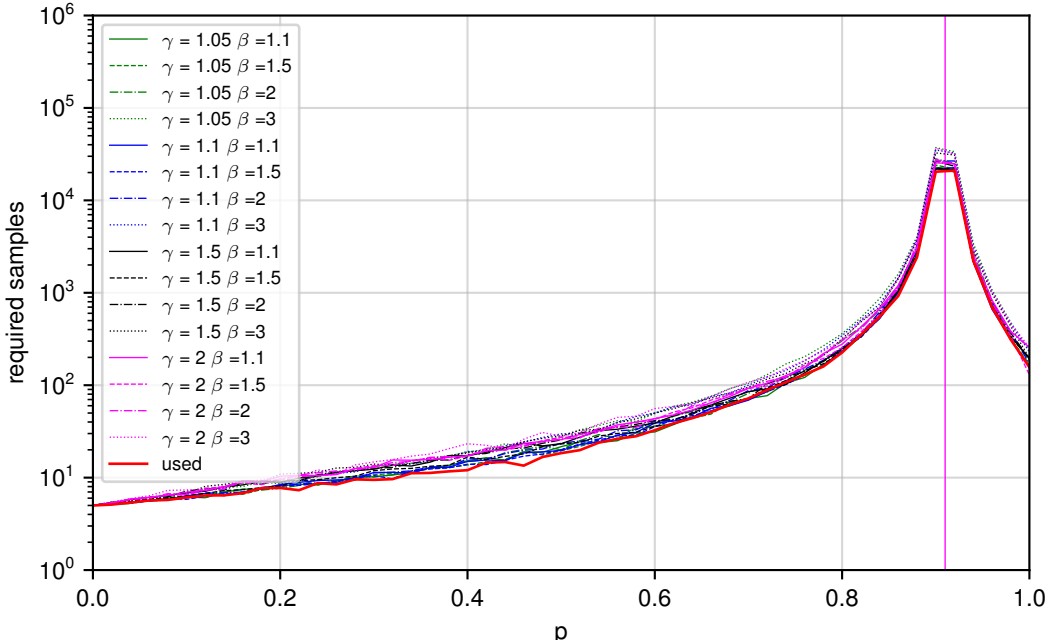

Figure 5: Samples needed for the adaptive estimation task as in Figure 2 for different hyperparameters. $\beta$ is the factor by which we enlarge the sample size before computing new confidence interval, $\gamma$ is the scaling of $\alpha$ as described in the main text. I.e., $k-$th estimation will have $\alpha_k = \frac{\alpha c}{k^\gamma}$ where $c$ is the normalization constant such that $\sum_{k=1}^{\infty} \alpha_k = \alpha$.

Figure 5. The width of the confidence interval scales as:

$$\sqrt{\frac{\beta(\log(1/(\gamma-1)) + \log(1/\alpha) + \gamma \log\log_\beta t)}{t}}.$$

This is because $\sum_{t=1}^{\infty} \frac{1}{t^\gamma} \asymp \frac{1}{\gamma-1}$, $\alpha_k \asymp \frac{\alpha(\gamma-1)}{k^\gamma}$ and $t \asymp \beta^k$, then $k \asymp \log_\beta t$. Plugging in these identites in $\sqrt{\frac{\log(\alpha_t)}{t}}$, and remembering that the confidence interval is recomputed only after elarging the sample size by $\beta$ factor, then for time $t$, the actual $t$ we use in the formula can be as small as $t/\beta$.

### C.2.1  Comparison with Horváth et al. (2022)

The method from Horváth et al. (2022) uses a finite collection of values of $n = n_1 < \cdots < n_s$ for which the confidence intervals are computed. A direct consequence is that the hard examples for which more than $n_s$ samples are needed cannot be certified. Additionally, due to Bonferroni correction, with large $s$, the term $\log(s/\alpha)$ appearing in the confidence interval becomes large compared to $t$ for small values of $t$ (compare with the polynomial scaling that we propose where this is not the case). Consider $n_i = n_1^i$, then the width of the confidence interval scales as (not considering the regime when $t > n_s$ where the width is constant):

$$\sqrt{\frac{n_1(\log s + \log 1/\alpha)}{t}}.$$

In particular, when we want to have confidence sequence up $n_s = N$ samples, then the width becomes

$$\sqrt{\frac{\sqrt[s]{N}(\log s + \log 1/\alpha)}{t}}$$

and we either pay for the fact that we have large differences between the steps ($\sqrt{s N}$), or for the fact that we have lot of steps ($\log(s)$) which is detrimental for small values of $t$.

## C.3 Details for 1, 3, 4

We had WideResNet-40-2 for CIFAR-10 trained for 120 epochs with SGD and learning rate $0.1$, Nesterov momentum $0.9$, weight decay $0.0001$ and cosine annealing. batch size 64. The loss was the standard, noise-augmented training using the same noise as for the certification. We either used Gaussian smoothing for $\ell_2$ robustness of uniform in $\ell_\infty$ box for $\ell_1$ robustness.

For the certification we used batch size 100 (natural for Horváth et al. (2022)). For our method, we had a mixed batch of data points so the data points for which we have the least amount of samples and are not decided yet are put in the batch.

## D Proof of Proposition 2.4

*Proof.* We show it for the upper interval, the lower is analogical. First, we note that $f(p) = \mathbb{P}(\mathcal{B}(n,p) \geq a)$ is non-decreasing in $p$ for any $n,a$; Thus, $u_r(X) \leq p$ if and only if $\mathbb{P}(\mathcal{B}(n,p) > x) + w\mathbb{P}(\mathcal{B}(n,p) = x) > \alpha$. Now, we show that $\mathbb{P}(u_r(X) \leq p) = 1 - \alpha$ for $X \sim \mathcal{B}(n,p)$. To shorten the notation, let $\alpha_1 = \mathbb{P}(X \geq a)$ and $\alpha_2 = \mathbb{P}(X > a)$ for such $a$ that $\alpha_1 \leq \alpha \leq \alpha_2$. We have

$$
\begin{aligned}
\mathbb{P}(u_r(X) \leq p) &= \mathbb{P}(u_r(X) \leq p \mid X < a)\mathbb{P}(X < a) \\
&\quad + \mathbb{P}(u_r(X) \leq p \mid X = a)\mathbb{P}(X = a) \\
&\quad + \mathbb{P}(u_r(X) \leq p \mid X > a)\mathbb{P}(X > a),
\end{aligned}
$$

which we evaluate to $\mathbb{P}(u_r(X) \leq p) = (1 - \alpha_1) + (\alpha_1 - \alpha_2)\mathbb{P}(u_r(X) \leq p \mid X = a) + 0$. We note that given the event $X = a$, it holds $u_r(X) \leq p \iff \alpha_2 + W(\alpha_1 - \alpha_2) > \alpha$, so $\mathbb{P}(u_r(X) \leq p \mid X = a) = \mathbb{P}(\alpha_2 + W(\alpha_1 - \alpha_2) > \alpha) = \mathbb{P}\left(W > \frac{\alpha - \alpha_2}{\alpha_1 - \alpha_2}\right) = \frac{\alpha_1 - \alpha}{\alpha_1 - \alpha_2}$. Overall, $\mathbb{P}(u_r(X) \leq p) = (1 - \alpha_1) + (\alpha_1 - \alpha_2)\frac{\alpha_1 - \alpha}{\alpha_1 - \alpha_2} = 1 - \alpha$.

The second part of the statement follows from Neymann-Pearson lemma. Concretely, we consider the following binary hypothesis testing problem from sample $\mathcal{B}(n,\theta)$, and we decide if $\theta = p$ or $\theta = q$. Both confidence intervals has size $\alpha$ and can be interpreted as binary tests – just return the indicator function of $q \in I(x)$. Neymann-Pearson lemma states that the (unique) uniformly most powerful test is the likelihood ratio test, which is implemented by the randomized Clopper-Pearson interval.

$\square$

## E Proof of Theorem 2.7

*Proof of Theorem 2.7.* First, $\sum_{i=1}^{\infty} \alpha_t = \sum_{k=1}^{\infty} \frac{\alpha}{k(k+1)} = \alpha$; thus, by union bound, all the computed confidence intervals are simultaneously correct at confidence level $1 - \alpha$. Next we show that the width is as claimed. When $t = 2^k$, we directly have from Hoeffding's inequality

$$
\varepsilon \lesssim \sqrt{\frac{\log \frac{1}{\alpha_t}}{t}} \asymp \sqrt{\frac{\log \frac{1}{\alpha} + \log \log t}{t}}.
$$

Otherwise, we would use a confidence interval of some previous $t'$ such that $t' < t < 2t'$ with width

$$
\varepsilon \lesssim \sqrt{\frac{\log \frac{1}{\alpha} + \log \log t'}{t'}} \asymp \sqrt{\frac{\log \frac{1}{\alpha} + \log \log t}{t}},
$$

Noting that Clopper-Pearson's confidence intreval is shorter than Hoeffding's finishes the proof. $\square$

## F Proof of Theorem 2.11

*Proof.* First we verify that everything in the algorithm is well defined and the logarithms take positive inputs. Now let $W_t = \exp\left(\frac{\text{LOGQ}_t}{\text{LOGP}_t}\right)$ where subscript $t$ denotes iteration of the algorithm. We show that it is a martingale when $X \sim \mathcal{B}(p)$ for $0 < p < 1$. In that case,

$$
\mathbb{E}_X[W_t] = W_{t-1}\mathbb{E}_X\left[\left(\frac{\hat{q}_t}{p}\right)^X \left(\frac{1 - \hat{q}_t}{1 - p}\right)^{1-X}\right] = W_{t-1}\left(p\frac{\hat{q}}{p} + (1 - p)\frac{1 - \hat{q}}{1 - p}\right) = W_{t-1}.
$$

|                            | r=0.5          | r=1.25 ,        | r=2           |
| -------------------------- | -------------- | --------------- | ------------- |
| Adaptive Horváth et al. (2022) | $1976 \pm 41$    | $3593 \pm 574$    | $4623 \pm 47$   |
| Betting CS 2               | $531 \pm 157$    | $2169 \pm 257$    | $2130 \pm 339$  |
| Union bound CS 1           | $635 \pm 157$    | $2557 \pm 234$    | $2670 \pm 271$  |
| Adaptive Horváth et al. (2022) | $0.13 \pm 0.006$s | $0.23 \pm 0.036$s | $0.3 \pm 0.003$s |
| Betting CS 2               | $0.05 \pm 0.012$s | $0.17 \pm 0.019$s | $0.17 \pm 0.02$s |
| Union bound CS 1           | $0.05 \pm 0.006$s | $0.19 \pm 0.018$s | $0.21 \pm 0.02$s |
|                            | r=0.5          | r=1.25 ,        | r=2           |
| Adaptive Horváth et al. (2022) | $4150 \pm 523$   | $2266 \pm 640$    | $4760 \pm 131$  |
| Betting CS 2               | $2206 \pm 150$   | $1665 \pm 84$     | $3932 \pm 199$  |
| Union bound CS 1           | $2665 \pm 78$    | $1674 \pm 37$     | $3717 \pm 361$  |
| Adaptive Horváth et al. (2022) | $0.27 \pm 0.04$s | $0.15 \pm 0.04$s  | $0.3 \pm 0.009$s |
| Betting CS 2               | $0.17 \pm 0.011$s | $0.13 \pm 0.006$s | $0.3 \pm 0.014$s |
| Union bound CS 1           | $0.2 \pm 0.005$s | $0.13 \pm 0.002$s | $0.28 \pm 0.02$s |

Table 3: Comparison of the average number of sample needed to decide if the point is certifiably robust with given radius in the top. The time needed is on the bottom. The upper table was in the main paper, the bottom one is the exact same experiment but with a retrained model for $\ell_2$ robustness robustness with $\sigma = 1$.

|                            | r=0.5           | r=1 ,           | r=1.5            |
| -------------------------- | --------------- | --------------- | ---------------- |
| Adaptive Horváth et al. (2022) | $423 \pm 38$      | $3520 \pm 82$     | $3823 \pm 127$     |
| Betting CS 2               | $138 \pm 28$      | $2680 \pm 221$    | $3007 \pm 122$     |
| Union bound CS 1           | $150 \pm 3$       | $2926 \pm 18$     | $3144 \pm 347$     |
| Adaptive Horváth et al. (2022) | $0.04 \pm 0.006$s | $0.23 \pm 0.006$s | $0.25 \pm 0.009$s  |
| Betting CS 2               | $0.19 \pm 0.002$s | $0.21 \pm 0.017$s | $0.23 \pm 0.01$s   |
| Union bound CS 1           | $0.016 \pm 0.006$s | $0.22 \pm 0.009$s | $0.25 \pm 0.03$s   |
|                            | r=0.5           | r=1 ,           | r=1.5            |
| Adaptive Horváth et al. (2022) | $1370 \pm 596$    | $4790 \pm 50$     | $8463 \pm 714$     |
| Betting CS 2               | $592 \pm 94$      | $4055 \pm 459$    | $5822 \pm 81$      |
| Union bound CS 1           | $806 \pm 113$     | $4327 \pm 106$    | $5795 \pm 321$     |
| Adaptive Horváth et al. (2022) | $0.1 \pm 0.04$s  | $0.30 \pm 0.003$s | $0.53 \pm 0.05$s   |
| Betting CS 2               | $0.05 \pm 0.007$s | $0.31 \pm 0.03$s  | $0.44 \pm 0.005$s  |
| Union bound CS 1           | $0.06 \pm 0.008$s | $0.33 \pm 0.008$s | $0.44 \pm 0.02$s   |

Table 4: Comparison of the average number of samples needed to decide if the point is certifiably robust with given radius in the top. The time needed is on the bottom. top and bottom are again the exact same experiment but with a retrained model for $\ell_1$ robustness with $\sigma = 1$.

If $p \in \{0, 1\}$, then we would have a deterministic sequence and $W_t \leq W_{t-1}$. Thus, $W_t$ is a supermartingale. It is also output of the exponential function and so is non-negative. Therefore, the assumptions of Ville's inequality are satisfied and can be applied. Whenever $p$ is excluded from the confidence interval, it happened that $\text{LOGQ}_t - \text{LOGP}_t \geq \log(1/\alpha)$, or equivalently, $W_t \geq 1/\alpha$ which can only happen with probability $\alpha$ and it is thus a valid confidence sequence. We also recall that in the main text we have shown that $I_p$ is a sub-level set of a convex function and is thus convex and can be efficiently found by binary search. The width of the confidence interval follows from the standard regret bounds for the Krichevsky–Trofimov estimator Cesa-Bianchi & Lugosi (2006) Section 9.7; Krichevsky & Trofimov (1981). The result then follows from Orabona (2019), Subsection 12.7. It was also derived in Ryu & Bhatt (2024); see Ryu & Wornell (2024) for generalization to vector-valued random variables.

$\square$

