# OpenReview forum: "Treatment of Statistical Estimation Problems in Randomized Smoothing for Adversarial Robustness"
_NeurIPS.cc/2024/Conference — NeurIPS 2024 poster_

### Official Review · Reviewer_xWqy · 2024-07-07

**Soundness:** 3
**Presentation:** 2
**Contribution:** 2
**Rating:** 5
**Confidence:** 4

**Summary:**

This paper presents two statistical innovations on top of standard randomized smoothing. The first is employing a randomized Clopper-Pearson interval (instead of deterministic), which marginally increases the certified radius for a particular number of samples. The second involves improving sample efficiency by leveraging confidence sequences, resulting in a roughly 2x speedup over existing methods.

**Strengths:**

I'm torn on this work -- although it seems to be a technically sound improvement on the SOTA, the method is very involved for a relatively small benefit. The paper could also use some better exposition. I'm curious what other reviewers think.

1. This work leverages interesting statistical techniques to improve upon randomized smoothing, both in terms of certified radius and sample efficiency.
2. The theory is quite extensive and seems sound (although I have not checked the proofs in detail).
3. Certified robustness of classifiers is an interesting and topical problem.

**Weaknesses:**

1. The authors should take care to proofread their work, which has a number of grammatical and structural errors.
2. The union-bound and betting-based confidence sequences seem to yield very similar results. I am not sure why both need to be included.
3. The paper flow is somewhat poor and difficult to follow. The authors should revise their manuscript to add better transitions between sections.
4. The proposed approaches include a great deal of complexity for relatively minimal improvement.

**Questions:**

1. Why is the blue line in Figure 1(a) so jagged?
2. I don't understand the purpose of the bounds mentioned on line 189: "In the analysis that follows, we assume that the Bernoulli distribution has a success probability satisfying 0 < c < p < C < 1 for some constants c, C, and thus we can hide the dependence on it into ≍."
3. I have trouble understanding the plots in Figure 2. The text of the paper suggests that epsilon is the width, so why is it plotted from 0 to 1 in the bottom left plot? What exactly are these bounds converging to? I feel that these plots are explaining something important but I'm not sure what the experimental setup is here exactly.
4. The randomized interval in Definition 2.3 only differs from the deterministic interval in the "knife edge" case where the binomial distribution is exactly x. I'd expect the contribution from this term to become negligible as the number of samples grows -- why is this not the case in Figure 2b?

**Limitations:**

The authors do not discuss limitations.

---

> ### Author Rebuttal · Authors · 2024-08-02
>
> Thanks a lot for your review. We appreciate the comments and questions that will help improve the clarity of the paper.  We would like to emphasize that the resulting (theoretically grounded) methods are reasonably simple ($\sim10$ lines of code) and (a heuristic) SotA uses $50\\%$ more samples and thus the improvement is significant. Furthermore, we provide lower bounds suggesting that it is impossible to significantly improve the performance estimation tasks.  We are happy to provide further clarifications if needed and we **kindly ask you to consider increasing the score** if you are satisfied with the responses
>
> ## Weaknesses
> 1. **Grammatical errors**
>
> Sorry for this. We are aware of some of them and will try to eliminate them.
>
> 2.  **Why are 2 methods included when they perform similarly?**
>
> We decided to include both methods for two main reasons apart from the completeness: (1) Betting  works better for easy problems while UB is better for the harder ones. (2) We do not only want to propose a method for randomized smoothing. We want the reader to understand the subtleties of the task so they can potentially use this framework in a different context. In certain settings it might be easier to adopt the UB-CS and elsewhere the betting one.
>
> 3. **Paper flow is difficult to follow**
>
> We tried to make the writing as clear as possible and we suspect that some difficulty is inherent due to the technical complexity of the presented topics. We would like to increase the clarity of the paper, could you please share in what parts were difficult to follow?
>
> 4.a  **Approaches include a great deal of complexity**
>
> The derivations of the methods are sometimes involved, but the methods themselves are then rather straightforward to implement. On line 243 (top of page 7) we show both proposed algorithms and they can be implemented in 12 simple lines of pseudocode (Python code would be in almost a 1-to-1 correspondence, invoking library function bisect). Union bound CS is just as complex code as the baseline (Horvath, 2022).
>
> 4.b **...for relatively minimal improvement**
>
> We want to stress that we consider the improvements of our methods to be significant. The SotA method (1) require roughly $50\\%$ more samples than our methods, (2) cannot be used in the very large sample regime and (3) is only heuristic
>
> On the contrary, our method is optimal in a certain sense and is competitive with ideal methods which strictly outperform the best possible methods for this task. Thus, we not only provide an empirically strong method outperforming SotA by $50\\%$, but also demonstrate that significant further improvements on this task are impossible.
>
> ## Questions
> 1. **Why is the blue line in Fig 1 jagged?**
>
> This is a property of the Copper-Pearson intervals - that we wanted to demonstrate - resulting in them being conservative in general. We discuss it in the paragraph starting at line 138. In Example A.2 (and figure 3) we work out a simple example for B(2,p) using elementary math demonstrating why this behavior occurs. Briefly, in setting of Fig 1a, $p_1=0.912$ and $p_2=0.933$ will **not** be contained in the upper confidence interval only when observing $100$ heads from $100$ tosses. This happens for $p_1$ with probability $\alpha_1 = 10^{-4}$ and for $p_2$ with probability $\alpha_2 = 10^{-3}$ which are then points on the jagged blue line.
>
> 2. **Why do we assume that there are some constants $c,C$ such that $0 <c <p <C < 1$?** (answer copied for  Reviewer NzaR)
>
> This is a purely technical assumption for the clarity of exposition. This way we can hide the dependency of the width of the confidence interval on $p$ (so the width scales as $n^{-\frac12}$ and thus simplify the discussion on the width when $np \asymp 1$ where it exhibits complicated behavior (anything between $n^{-\frac12}$ and $n^{-1}$) covered in the referred book. We move the discussion in appendix and replace the paragraph in the main text by the following:
>
> For the simplicity of exposition, let the width of a confidence interval at level $1-\alpha$ with $n$ samples be $\asymp \sqrt{\log(1/\alpha)/n}$. This way, we hide the dependency on $p$ into $\asymp$. In the full generality, the width of the confidence intervals exhibits many decay regimes between the rates $\sqrt{p(1-p)\log (1/\alpha)/n} $ (when $np \gtrsim 1$) and $\log(1/\alpha)/n$ (when $np \asymp 1$).  Our algorithms capture the correct scaling of the confidence intervals. Further discussion is provided in Appendix.
>
> 3. **Understanding Fig 2**
>
> Our bottom-left y axis label might have not been the best choice, we will update both captions on the left by words. In the notation of Algorithm 1,2; in the top-left we plot U_t - L_t, while on bottom-left we plot both U_t and L_t (i.e., top = width of CS, bottom = CS itself). We find the setting to be described exhaustively in the caption, but we needed to keep it short due to space constraints. We will put the following in the caption of the left part: "In the notation of Algorithms 1,2, the sequence of $U-L$ is in the top figure, while both sequences $U$ and $L$ are in the bottom figure." See the enclosed pdf for the new figure.
>
> 4. **Why are the gains from randomized CI not diminishing with increasing $n$?**
>
> The randomized and deterministic CIs are only equal when the realization of a uniform r.v. on the interval [0,1]  is 1 (so almost surely the randomized ones are larger). The difference in widths gets smaller as $n$ grows; however in Fig 1b. we show the certified radii with Gaussian smoothing and so we plot $\Phi^{-1}(\underline p)$ ($\Phi$ is Gaussian CDF). It holds that $\lim_{p \to 1} \Phi^{-1}( p) = \infty $, so even when the absolute difference between the deterministic and randomized CI lower-bound  of $p$ decreases, the difference between the certified radii roughly stays the same because the growth of $\Phi^{-1}$  is increasing quickly around $1$. This holds only when $\underline p \sim 1$ which is the relevant part of Fig 1b.

---

> > ### Comment · Reviewer_xWqy · 2024-08-10
> >
> > Thank you to the reviewers for their clarifications. I'm raising my score to a 5, but am still not confident on this paper. Mostly I'm not sure that a 50% reduction in samples is significant, as randomized smoothing is very far from being practical in any sense without an order-of-magnitude conceptual breakthrough. This seems unlikely given that these approaches have been explored for a few years now.
> >
> > While this paper doesn't particularly excite me, perhaps the AC's tastes are different in this regard.
> >
> > Also as a small note: I recommend adding periods after all inline subsections (e.g., after "Related Work" in line 181).

---

> > > ### Author Response · Authors · 2024-08-10
> > >
> > > Thanks a lot!
> > >
> > > If you think that RS is impractical at this stage, then you can see our paper as a negative result, since our lower bounds prevent significant improvements for the considered task.
> > >
> > > On the other hand, we provide an adaptive estimation procedure that can be used for virtually any task related to randomized smoothing estimation and we can draw samples one-by-one.
> > >
> > > Thanks again for your review, we will add the commas.

---

### Official Review · Reviewer_NzaR · 2024-07-12

**Soundness:** 3
**Presentation:** 2
**Contribution:** 3
**Rating:** 7
**Confidence:** 4

**Summary:**

This paper proposes sample-efficient methods for computing probabilistic robustness certificates for randomized smoothing. The proposed methods replace the standard Clopper-Pearson confidence interval on the classifier’s score, with a confidence sequence, thereby allowing the number of samples to be determined adaptively given a radius of certification $r$ and significance level $\alpha$. Two variants of the method are proposed: one that updates the sequence on a geometric schedule (shown to achieve the asymptotically optimal sample complexity) and another that adopts a betting strategy. The methods are shown to consume 1.5–3 times fewer samples compared to prior work (Horváth et al., 2022).

**Strengths:**

**Originality:**
This paper imports methods from statistics/probability theory, which have not previously been applied in the context of randomized smoothing. The methods appear to be an excellent fit for improving the statistical efficiency (and hence computational efficiency) of randomized smoothing. It’s worth noting that the methods have been adapted, in that the bounds are specialized for Bernoulli random variables (rather than generic bounded random variables).

**Significance:**
The computational cost of randomized smoothing is a barrier to adoption, so it’s great to see work in this direction. The proposed algorithms for adaptively determining the sample size is relatively simple to implement which should encourage adoption.

**Quality:**
The method is well-motivated. The theory and experiments are generally well-executed, apart from some minor issues outlined below.

**Clarity:**
The writing is reasonably clear, apart from some issues discussed below.

**Weaknesses:**

**Clarity:**
The paper is mostly clear, however some parts could be improved:
- Section 3: Definition 3.1 seems out-of-place: it’s not clear that a generic formulation of sequential decision making is needed, if the experiments focus on randomized smoothing. Section 3.1 includes some discussion of the results, but is labeled “related work”. It would be good to add a few more sentences discussing the results
- Section 2.2: I find it confusing that symmetric/asymmetric confidence sequences are discussed before confidence sequences are defined (even informally).
- Line 189: I found this paragraph confusing – I wonder if it could be explained more concretely (with an example) in an appendix.

**Impact:**
- The formulation of randomized smoothing in Section 2 covers additive smoothing mechanisms, where the certificate is an $\ell_p$-ball. I suspect the formulation could be generalized to capture non-additive smoothing mechanisms and more general certificate geometries without impacting the validity of the results.
- The proposed method is not applicable if one wants to estimate the maximum certified radius at an input. I wonder whether the method could be adapted to estimate the maximum certified radius within some tolerance.

**Minor:**
- line 24: Provide citation for claim that randomized smoothing “is currently the strongest certification method”
- line 62: “realizations **are** lowercase”
- line 77: Extraneous closing bracket
- line 88: “de-randomized” has not been defined yet. Consider defining earlier or providing a citation.
- line 92: Provide citation for claim that the Clopper-Pearson interval is “well known to be conservative”
- line 94: In what sense is the confidence interval “optimal”? In terms of coverage?
- line 121: “even” → “event”
- line 128: Is the case where u() = 0 an upper confidence interval?
- Proposition 2.4: Coverage has not been defined for a randomized confidence interval.
- line 180: Should $\in$ be $\subseteq$?
- line 247: Provide citation for Ville’s inequality
- Table 1 caption: Delete “of number of samples”
- line 324: “improved them at places” is ambiguous.
- line 328: I’m not convinced that we now have a “perfect” understanding of statistical estimation for randomized smoothing. For instance, this paper has not considered the problem of estimating the smoothed classifier’s prediction. It’s possible there may be more sample-efficient ways of estimating the smoothed classifier’s prediction and bounds on the top-2 scores jointly.
- “We stress out <some statement>” sounds unusual. It would be more natural to say “We stress <some statement>”.

**Questions:**

- Does the method apply to the more general formulations of randomized smoothing?
- Could the method be adapted to estimate the maximum certified radius within some tolerance? This would cover another common use case.

**Limitations:**

The limitations ought to be discussed more. For example, the experiments only cover one dataset/model, the proposed method only improves sample complexity if the radius is fixed.

---

> ### Author Rebuttal · Authors · 2024-08-03
>
> Thanks a lot for your thorough review! We appreciate the insightful comments that will help with the paper.
> If we satisfactorily answer your questions and reservations, we kindly ask you to consider increasing your score.
>
> ## Questions
>
> * **Does it apply to more general formulations of smoothing?**
>
> Yes, our methodology can be directly applied to all[1] randomized smoothing works we are aware of as they have the same estimation subroutine. Some examples of such non-additive smoothing are Wasserstein smoothing[2] and Image transformation smoothing[3] and they both use the standard estimation procedure in randomized smoothing due to (Cohen, 2019). We state this explicitly in the paper
>
> [1]: modulo the deterministic ones, where there is no estimation. Additionally, rarely a soft base classifiers are considered instead of hard ones. Here one would need to use e.g., empirical Bernstein bound instead of Clopper-Pearson; alternatively the betting wealth needs to be computed in a different way. \
> [2]: Levine et al. Wasserstein Smoothing: Certified Robustness against Wasserstein Adversarial Attacks\
> [3]: Fischer et al. Certified Defense to Image Transformations via Randomized Smoothing
>
> * **Applicability to more general tasks; e.g., estimate the maximum certified radius within some tolerance**
>
> Yes, the methods can be directly applied. We run the confidence sequences until a stopping condition is met (in the paper, it was that certain value of $p$ is outside of the confidence sequence). In the proposed task the stopping criterion might be that $r(hi) < r(lo) + \varepsilon$, where the current confidence interval on the probability is $[lo, hi]$ and $r(p)$ is the radius computed from probability $p$. In the example, we know (at certain confidence) that $r(lo)$ is underestimating the true certified radius and $r(hi)$ is overestimating it; and so when the stopping criterion is met, we return conservative estimate $r(lo)$, but we know that it is at-most $\varepsilon$ smaller than the true radius. We will include this task in the paper. Please see the enclosed pdf for preliminary results.
>
> ## Clarity
> Thanks for the pointers, we will fix the problems.
>
> * **Start of setction 2.2**
>
> We move the content of the first paragraph after the definition 2.5 and the following remark.
>
> * **Section 3, definition 3.1**
>
> We agree that the section is brief, we will expand it using the extra page. In the paper, we try to strike a balance between the generality (so the results can be readily transfered) and the clarity of the connection to randomized smoothing. We think that the task from Definition 3.1  is natural and we would like to keep it in this form. We agree that the definition might seem to come out of nowhere and we will provide more motivation for it.
>
> * **Paragraph after line 189**   (answer copied for Reviewer xWqy)
>
> This is a purely technical assumption for the clarity of exposition. This way we can hide the dependency of the width of the confidence interval on $p$ (so the width scales as $n^{-\frac12}$ and thus simplify the discussion on the width when $np \asymp 1$ where it exhibits complicated behavior (anything between $n^{-\frac12}$ and $n^{-1}$) covered in the referred book. We move the discussion in appendix and replace the paragraph in the main text by the following:
>
> For the simplicity of exposition, let the width of a confidence interval at level $1-\alpha$ with $n$ samples be $\asymp \sqrt{\log(1/\alpha)/n}$. This way, we hide the dependency on $p$ into $\asymp$. In the full generality, the width of the confidence intervals exhibits many decay regimes between the rates $\sqrt{p(1-p)\log (1/\alpha)/n} $ (when $np \gtrsim 1$) and $\log(1/\alpha)/n$ (when $np \asymp 1$).  Our algorithms capture the correct scaling of the confidence intervals. Further discussion is provided in Appendix.
>
> ## Minor
> Thanks for all the relevant points. We integrate all of them but comment here only one the "open ended" ones.
>
> * **In which sense is the randomized interval optimal?**
>
> They are the shortest possible in expectation. We state it formally in the paper: For any binomial r.v. $X \sim \mathcal{B}(n, p)$ and any $q$ (where $p,q$ are probabilities and $n$ is number of samples), $\mathbb{P}(q \in CI_{\text{our}}(X)) \leq \mathbb{P}(q \in CI_{\text{other}}(X)) $, where $CI_\text{our}$ is our proposed randomized confidence interval and $CI_\text{other}$ is any other confidence interval  at the same confidence level as ours, and so our intervals are the shortest one in expectation.
>
> * **Is [0, v(x)] the upper confidence interval?**
>
> We called it the lower confidence interval since it contains the low values. On the other hand $v(x)$ is the upper bound for the estimated quantity. We are open to rename the interval if it helps the clarity.
>
> * **Definition of coverage for randomized intervals**
>
> We update the definition so to cover the random intervals. The coverage is the probability of the true parameter appearing in the confidence interval, where the probability is not only over sampling, but also over the randomness of the intervals.
>
> * **Problems with Conclusions**
>
> Thanks, we agree. We rewrite it in the following way:
>
> In this paper, we investigated the statistical estimation procedures related to randomized smoothing and improved them in the following two ways: (1) We have provided a strictly stronger version of confidence intervals than the Clopper-Pearson confidence interval. (2) We have developed confidence sequences for sequential estimation in the framework of randomized smoothing, which will greatly reduce the number of samples needed for adaptive estimation tasks. Additionally, we provided matching algorithmic upper bounds with problem lower bounds for the relevant statistical estimation task.
>
> ## Limitations
>
> We will add Imagenet experiments and with multiple models (we already have multiple models for CIFAR and $\ell_1, \ell_2$ tasks.) We will include your proposed task.

---

> > ### Comment · Reviewer_NzaR · 2024-08-08
> >
> > The authors' responses to my two questions have clarified that the proposed methods have broad applicability in randomized smoothing. I have therefore increased my score for "Contribution" and the overall rating.
> >
> > > Yes, our methodology can be directly applied to all[1] randomized smoothing works we are aware of as they have the same estimation subroutine. ... We state this explicitly in the paper.
> >
> > This was not clear to me from reading the paper. In section 2, randomized smoothing is formulated for additive noise and metric balls induced by a norm. Perhaps the authors could include a remark that this more limited formulation is used for ease of exposition, noting that the methods apply more generally as discussed in the response above.

---

> > > ### Author Response · Authors · 2024-08-08
> > >
> > > Thank you! We will remark it.

---

### Official Review · Reviewer_mvjJ · 2024-07-12

**Soundness:** 3
**Presentation:** 3
**Contribution:** 3
**Rating:** 6
**Confidence:** 2

**Summary:**

They study the task of certified robustness, i.e. they need to decide if a point is robust at a certain radius or not, using as few samples as possible while maintaining statistical guarantees. Their main contribution is utilizing confidence sequences (instead of confidence intervals) that allows them to draw just enough samples to certify robustness of a points which allows them to greatly decrease the number of samples needed. They also show the effectiveness of their approach experimentally.
Beyond that, they propose a randomized version of Clopper-Pearson confidence intervals for estimating the class probabilities. A standard component of randomized smoothing procedures is the Clopper-Pearsons confidence interval which is known to be conservative. As a result, the certification procedures underestimate the certified robustness. They provide an optimal confidence interval for binomial random variables that resolves this issue.

**Strengths:**

Certainly one of the main issues of Randomized smoothing methods is that they are not practical due to their computational burden, i.e. a lot of samples need to be drawn to decide robustness at a certain radius. Given that, their result seems interesting.

**Weaknesses:**

Other suggestions:

Line 118: hence instead of whence

Line 121: event instead of even

Contributions paragraph: Perhaps in the first paragraph mention that your results hold for binomial random variables.

**Questions:**

same as weaknesses.

**Limitations:**

same as weaknesses.

---

> ### Author Rebuttal · Authors · 2024-08-02
>
> Thanks for your review and typos! We will clarify the binomial case. We are ready to answer any questions if any arise!

---

> > ### Comment · Reviewer_mvjJ · 2024-08-12
> >
> > I went through other reviews and responses. Due to the computational cost of RS it is hard to adopt them, and they show an empirical method outperforming SotA by 50% reduction in samples. But it is true that RS is very expensive and it is not clear that 50% reduction in the number of samples is good enough. Furthermore, they show lower bounds that significant further improvements on this task are impossible. I believe the contribution is sufficient enough for acceptance.

---

> > > ### Author Response · Authors · 2024-08-12
> > >
> > > Thanks!

---

### Author Rebuttal · Authors · 2024-08-07

We thank the reviewers for their reviews! In general we agree with them. In our understanding, the reviewers agree on the fact that we successfully attacked a well-known limitation of randomized smoothing. Their perceived weaknesses are occasional writing problems - those are easily fixable and do not require significant changes.

In the enclosed pdf we provide updated figure 2. for better clarity. We changed the Y-axis labels on the left part. We also slightly updated caption -  changes are in red.

We also enclose a table with an experiment on a new task as requested by Reviewer NzAR

---

### Decision · Program_Chairs · 2024-09-25

**Decision:**

Accept (poster)

**Comment:**

This paper studies the statistical cost of certification of robustness based on randomized smoothing (RS). The certification step is one of the computational bottlenecks of RS and, as such, providing procedures that can reduce the amount of samples needed to certify a certain level of robustness at a given point is important. The authors provide a testing procedure based on confidence sequences and testing by betting that provides comparable certificates to the alternatives with about half of the needed samples. The manuscript also provides a lower bound demonstrating that further significant improvements in sample complexity will not be possible. There is a consensus among reviewers that this is a solid contribution, and I agree.